# An exploratory study of associations between judgement bias, demographic and behavioural characteristics, and detection task performance in medical detection dogs

Sharyn Bistre Dabbah[1], Michael Mendl[1], Claire Guest[2], Nicola J. Rooney[1,2]*

**1** University of Bristol, Langford House, Langford, Bristol, United Kingdom, **2** Medical Detection Dogs, Great Horwood, Milton Keynes, United Kingdom

* nicola.rooney@bristol.ac.uk

## Abstract

Medical detection dogs search for diseases from remote samples (biodetection) and assist patients with chronic conditions (medical alert assistance). There is scarce information on how dogs' decision-making tendencies relate to task performance. This study explored the relationships between medical detection dog demographics, responses in a behavioural test battery, 'optimistic' or 'pessimistic' decisions in a judgement bias task, and their performance in detection tasks. A sample of 58 trainee and trained medical detection dogs were studied in a Go/NoGo spatial judgement bias test. For trainee dogs (n=39), training outcome (pass/fail) and trainer ratings of behavioural traits; yielding a composite score of ability in detection tasks, were used as markers of task performance. For trained biodetection dogs (n=27), scent sensitivity and specificity scores derived during training and testing trials were used. Older dogs (p <0.001), those showing higher 'Confidence' (p=0.009), 'Food orientation' (p=0.014) and 'Playfulness' (p=0.005) in the test battery, and those who made more 'optimistic' decisions in the judgement bias task (p=0.002), had higher detection task ability scores. For trained dogs, latency to approach ambiguous stimuli was positively correlated with scent specificity levels (n=25, p=0.021), suggesting that more 'pessimistic' dogs tended to be more specific. Our findings suggest relationships between behaviour in judgement bias tests and other learning and discrimination tasks, which may reflect underlying individual or personality differences in affective and/or cognitive processes that influence dogs' style of searching and performance ability in medical detection tasks. Future research is needed to explore these associations further and investigate the value of judgement bias tasks in predicting later search performance in medical and other types of search dogs.

## Introduction

Judgement (or cognitive) bias tests (JBT) are commonly used in animal studies to assess the association between affective states and decision-making under ambiguity, and to use decision-making as a marker of animal emotion. Subjects learn to make a 'positive' response (e.g.,

**Data availability statement:** All relevant data for this study are publicly available from the University of Bristol data repository (https://doi.org/10.5523/bris.3hl3of08pc6432hubhr8nwesgj).

**Funding:** The current study was supported by a doctoral scholarship awarded to SBD from the Council of Science and Technology. MX (Consejo Nacional de Ciencia y Tecnología, CONACyT) (reference number: 472257; https://conahcyt.mx/) In partnership with the University of Bristol, UK (https://www.bristol.ac.uk/). The funders had no role in study design, data collection and analysis, decision to publish, or preparation of the manuscript.

**Competing interests:** I have read the journal's policy and the authors of this manuscript have the following competing interests: CG and NR are employed by Medical Detection Dogs, the charity where the dogs from this study are trained and data collection was conducted. However, they were not involved in data collection or analysis for this study. This does not alter our adherence to PLOS ONE policies on sharing data and materials.

approach) to a cue (e.g., a tone or location) predicting a positive outcome (e.g., food), and a 'negative' response (don't approach) to another cue (different tone or location) predicting a less positive outcome (no food) [1]. Once trained, they are presented with occasional ambiguous cues (e.g., intermediate tones or locations) and their responses recorded. In humans, positive judgements of ambiguity, often referred to as 'optimistic', are associated with positive affective states and negative judgements ('pessimistic') with negative states [1–4]. In non-human animals, there is evidence from many studies, including those on dogs [e.g., 5,6,7], that positive (labelled 'optimistic') or negative ('pessimistic') responses to ambiguous cues can be used as markers of positive or negative affective states respectively (for meta-analyses, see: [8,9]).

JBTs have been used as a marker of affective state in studies of dogs experiencing situations likely to generate negative states, finding, for example, that separation-related behaviour is associated with a 'pessimistic' judgement bias [10,11], as is training using aversive methods [12]. There is also evidence that, in dogs with separation-related behaviours, pharmacological treatment can reverse 'pessimistic' decision-making under ambiguity [13], and that more 'optimistic' decisions are observed in dogs administered oxytocin, which is thought to induce a relatively positive state [14], and after an olfactory enrichment programme [15,16]. A recent study investigated the association between a dog JBT and the scent of stress in non-familiar humans. After three sessions, dogs were significantly less likely to approach an ambiguous JBT location when exposed to odour from stressed humans compared to that from relaxed people, suggesting that odours produced by stressed humans may influence a dog's affective state and learning ability [17]. Other studies have investigated associations between JBT responses and chronic health issues including osteoarthritis, idiopathic epilepsy [18] and syringomyelia in Cavalier King Charles Spaniels [19]. Overall, there is evidence that decision-making under ambiguity may be a useful indicator of affective state in dogs, although predictions are not always confirmed [e.g., 11,18].

Other studies have investigated the relationship between JBT and non-affective variables including dog behaviour [16], age [20], and cognitive measures such as spatial memory [21] and laterality [22]. The vast majority of these studies have been on pet dogs but, to our knowledge, JBTs have not been used in working dogs, such as medical detection dogs. Therefore, this study aimed to gather JBT data in this population.

Detection dogs are trained to find an extensive range of items. [23–28]. Medical detection dogs (MDDs) can identify volatile organic compounds from a range of clinical conditions [29,30]. Some animals spontaneously acquire this skill [e.g., 31], and training can enhance it [e.g., 29,32]. MDDs undertake various roles. Biodetection dogs operate remotely under controlled conditions and are trained to differentiate between scent samples from diseased and healthy individuals. Medical alert assistance dogs (assistance), such as glycaemia alert dogs, work closely with a patient and notify them before an impending crisis occurs [33].

One way of quantifying biodetection MDD performance is in terms of scent detection sensitivity and specificity measured objectively during *in vitro* trials or real-life searches. In signal detection terms [34], the target scent is considered a signal (S+ condition), while other stimuli (non-target scents) are considered noise (S- condition). Dogs are trained to alert to the presence of S+ and to ignore the presence of S- [See 35]. When the dog correctly indicates an S+, it is referred to as a 'Hit' (true positive), but incorrect alerts to S- are 'False alarms' (false positives). If the dog correctly ignores an S-, it is referred to as a true negative, but if it misses an S+, it is a 'Miss' (false negative) [32,35,36].

Scent sensitivity refers to the number of true positives divided by the total number of S+ presentations [36]. It indicates how effective the dog is at detecting a target scent correctly.

$$\text{Sensitivity} = \frac{\text{True positives}}{\text{True positives} + \text{False negatives}}$$

Scent specificity refers to the number of true negatives divided by the total number of S- presentations. It shows the dog's ability to ignore the noise (e.g., blank samples or control scents) [36].

$$\text{Specificity} = \frac{\text{True negatives}}{\text{True negatives} + \text{False positives}}$$

Both high sensitivity and specificity are necessary for good detection performance, but the optimal balance will depend on the task's requirements [37,38].

The charity Medical Detection Dogs® [39], located in Milton Keynes, UK, where this research was mainly conducted, trains biodetection dogs and assistance dogs. During biodetection training, scent samples are presented to the dogs in a line or circle arrangement [40], interspersing S + among S- samples. Dogs are reinforced when showing a trained response to S + [e.g., 41, e.g., 42]. Training difficulty gradually increases by reducing the number of S + [e.g., 42] and increasing the array of confounding scents the dog is trained to ignore [e.g., 41]. S- (blank lines with no target) are also trained and included in testing. Due to randomisation, a dog may encounter two consecutive S-. In such cases, the response to repeated S- results in a non-reward, as there are no targets. Testing frequently involves a double-blind procedure where the trial's composition is unknown to all participants [37], from which sensitivity and specificity data are generated. Detection of some scents, e.g., COVID-19, may require the dog to search people or an outdoor environment, with the goal of eventually working in, for example, hospitalised patients or at a port of entry [e.g., 43,44]. Assistance dog training starts with *in vitro* trials with S + (from the prospective client), and subsequently, the dog is conditioned to identify S + on a person. The dog is trained to alert to the scent with specific behaviours, e.g., sitting or, in some cases, retrieving a blood testing kit, although it may perform additional behaviours to attract the client's attention [45].

Detection dogs vary in their sensitivity and specificity [e.g., 37,40,42,46] and, we hypothesise this in turn, could be related to inherent differences in their 'optimism' or 'pessimism' – indeed JBT studies in a number of species have identified stable individual differences in these characteristics [e.g., 47–51]. Dogs performing more 'optimistically' in a JBT may also be those who tend to be more sensitive 'liberal' detectors with an elevated estimation of the likelihood of the rewarded target scent and hence an elevated alert rate to increase the chance of a 'hit' (true positive) [36,38]. They may thus minimise false negatives, but may also have a higher tendency for false alarms. Conversely, dogs who make more 'pessimistic' choices in a JBT may have an elevated estimation of a non-rewarded outcome and, hence, also be more specific 'conservative' detectors tending to avoid false positives but also potentially missing targets [36,38].

Tangential evidence for these potential relationships comes from a study of diabetes alert detection dogs in which 'Willingness to try new behaviours' was reported to be linked with more accurate alerting of out-of-range events in clients with type I diabetes [52]. In humans, the tendency to take the initiative and try different strategies to achieve goals has also been associated with positive achievements [53]. It is thus possible that dogs that tend to judge ambiguous situations 'optimistically' might also be more proactive in trying new things to achieve a reward than more 'pessimistic' dogs. If such relationships between JBT and detection performance exist, JBTs may be useful in better understanding and predicting decision-making inclinations in detection dogs including MDDs.

To investigate this possibility, the first step is to establish whether there is indeed a cross-context correlation between performance in JBTs and detection tasks, and this was one aim of the current study. To this end, we assessed the association between dogs' performance in a JBT and measures of their success in MDD tasks including training

programme outcome (pass or fail) and a composite overall ability measure derived from the trainer's ratings of their dogs' performance. We also examined the link between JBT performance, dog demography, and behavioural characteristics measured in a test battery from a previous study [54]. Specifically, we hypothesised that dogs that behaved more 'optimistically' in a JBT would also be more sensitive detectors, whilst those that behaved more 'pessimistically' would be more specific detectors. A spatial version of the JBT was used.

### The judgement bias paradigm

In the spatial version of this task dogs are trained to distinguish between a positive baited bowl location and negative unbaited location over a number of trials and are then presented with a series of trials in which bowls are placed in intermediate positions. Relative running speeds to these locations are taken as measures of 'optimism' vs 'pessimism'. In the original dog study using this approach [5], the handler led the dog behind a visual barrier so that it couldn't see whether the tester baited the bowl on each trial. However, restraining the dog behind the barrier could be stressful or frustrating for the dog. Therefore, Hale [55] designed a wooden screen that was placed just behind the bowl locations, allowing the tester to bait (or not bait) the bowl out of the dog's sight and then slide the bowl through holes in the board at specific locations. The revised experimental setup also shortened the time to prepare each test trial. Hence, the present study used the same apparatus and methods adapted from Hale [55].

## Materials and methods

### Ethical statement

The current study was approved by the University of Bristol (Animal Welfare and Ethics Review Body (AWERB) Ref UB/19/05). During the study, the trainers handled the dogs and provided information about them, rating various aspects of their training performance. However, the assessments and reports on the dogs were considered part of the staff regular duties, and because the Medical Detection Dogs (MDD) organisation requires staff to assess and test their dogs and this data was made available, and no trainers' personal information was requested no ethical approval for human participation was necessary. However, handlers gave verbal consent for their data to be used in the study.

### Experimental sample

The sample included all prospective medical detection dogs over 12 months old that were being socialised or trained from 09/09/2019 to 10/03/2020 (N = 39) and all dogs working on biodetection dog projects (N = 19) during that period. This totalled 58 MDDs.

Half of the dogs were female (N = 29), 86.2% of males were neutered (N = 25), and 82.8% of females were spayed (N = 24) and their mean age was 32.3 (±26.38) months (min = 11, max = 132). Breeds included 31 Labrador Retrievers, 10 Labrador/Golden Retriever cross, 8 Cocker Spaniels, 4 Golden Retrievers, 2 Labradoodles, a Border Collie, an English Springer Spaniel and a Hungarian Wire-haired Vizsla. Thirty-two dogs came from breeders procured as puppies, 23 from other assistance dog charities, and three from dog rehoming charities.

Of the 39 trainee dogs, at the test point, 28 were being socialised, and 11 were in initial biodetection or alert assistance training. However, by the end of data collection (19/06/2021), an additional eight dogs were working on biodetection tasks, and their scores were used for the subanalyses on biodetection dogs. Dog details can be found in Supporting Information (S1 Table).

## Experimental setting

The judgement bias task was conducted in a room in the Biodetection wing of the Medical Detection Dogs® charity training facility (Fig 1). The test was video recorded with a Swann® CCTV system. The room measured 6.80m x 3.7m, was partially carpeted and had a smaller tiled section (2.3m²). The entrance door to the room was located centrally along one wall. There were two windows in the opposite wall. A table was at the tiled end. A 3.5m wooden screen with three panels used for the JBT was at the opposite end. The room temperature was set between 21-22 Cº to avoid discomfort to the dogs from weather fluctuations. A bowl with fresh water was available all the time.

The wooden screen structure [55] was 3m wide and consisted of three panels with 25 cm wide doors for each of the five possible bowl locations (Figs 1 and 2). Each door opened backwards via a small fabric handle.

During training the negative (N) and positive (P) locations were at the far ends of the screen (Fig 2). Their locations were counterbalanced across the dogs. The 'Middle location' (M) was in the central panel, the 'near positive' (NP) was by P, and the 'near negative' (NN) was by N. Each bowl location was marked 20 cm in front of the door with a discrete X on the carpet to indicate where the tester had to place the bowl.

The tester remained behind the screen during the whole procedure and was therefore not visible to the dogs. The CCTV system (Swann®) mounted on the ceiling pointing towards the screen allowed the tester to observe the dogs on a monitor, recording the dogs' latencies to reach the bowls in real time. Another CCTV camera mounted to the ceiling in the middle of the room recorded a central room view, and a Canon® video camera was placed on the table

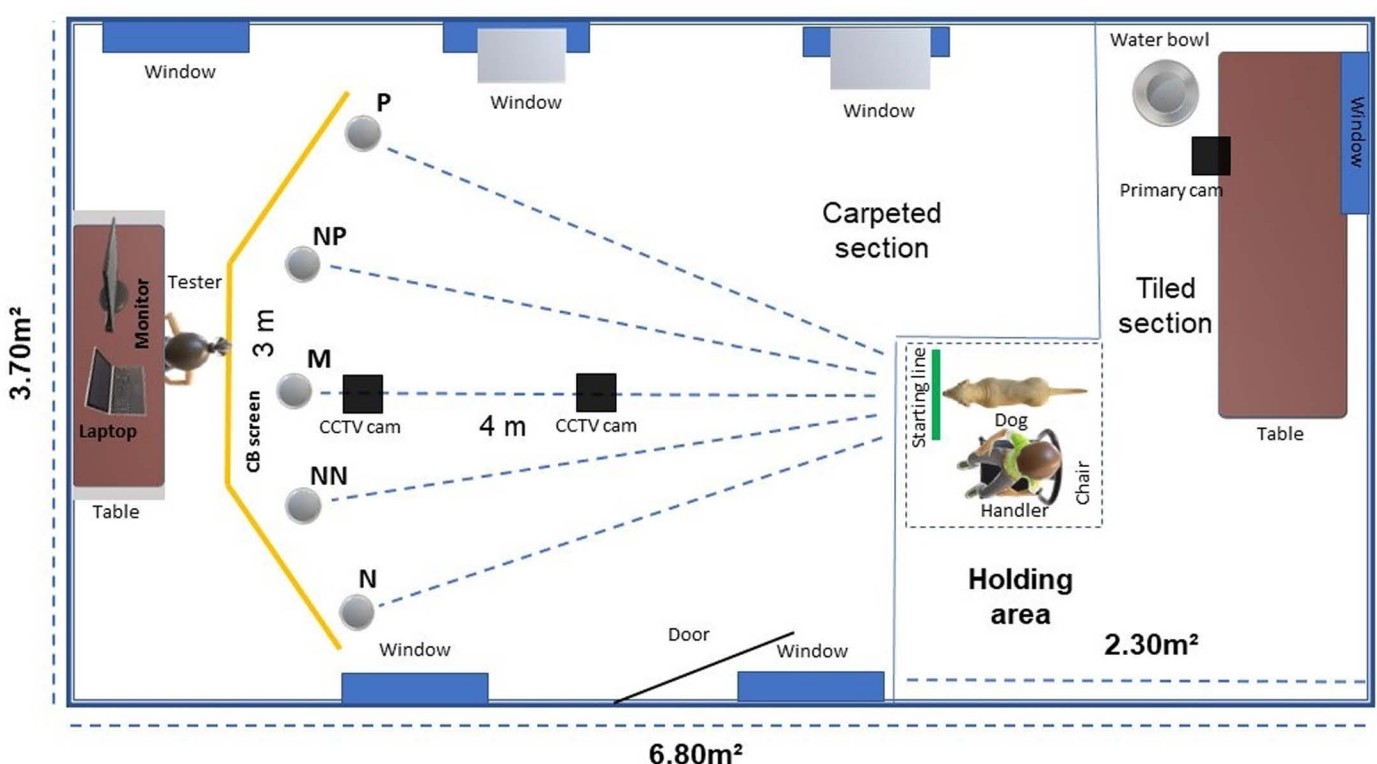

**Fig 1. Overhead view of the experimental room.** Shows the JBT arrangement with room measurements, the items utilised in the JBT and their location, and the dog's initial position. Cam=Camera. Bowl locations on JB screen: P = Positive; NP = Near positive; M = Middle; NN=Near negative; N = Negative.

at the opposite end. The handler (dog trainer) and dog waited in the holding area 4m from the test apparatus before each trial (Figs 1 and 2).

## The JBT procedure

The dogs (N = 58) were first trained to discriminate between positive and negative locations before being tested with ambiguous locations. JBT always took place after a battery of other tests, briefly summarised in the Methods and Supporting Information (S2 Table), which took approximately 90 mins to complete and from which behavioural variables were generated for use in the current study [54]. There was a minimum of 120 minutes of rest (mean 123 ±

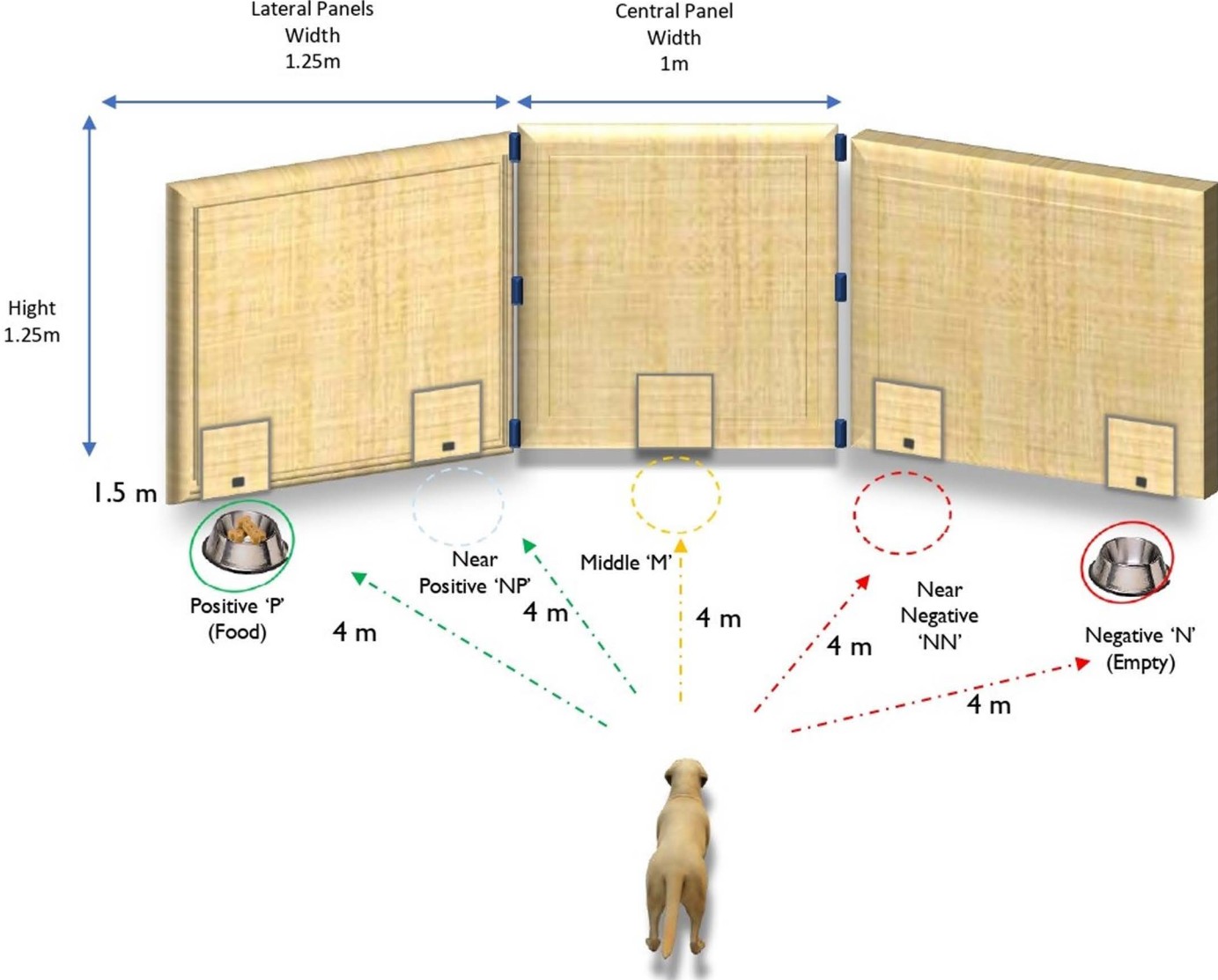

**Fig 2. Front view of the JBT apparatus containing five possible bowl locations.** The positive 'P' (marked in green) and negative N' (marked in red) locations on either side of the wooden panel were the conditioned locations. The other three positions in between those locations (near positive, middle and near negative) presented ambiguous test locations.

1.6, max = 132) between the end of the test battery and the start of the JBT to ensure a break between tests. The JBT lasted 52.4 ( ± 1.3) minutes on average (min = 45.2, max = 58.2).

On arrival, dogs were allowed to freely explore the room for 5 to 8 minutes while the tester explained the test procedure to the handler (the dog's trainer). However, animals were already habituated to the room since it was also used during the test battery.

The tester (SBD) then went behind the screen where she remained until the end of the procedure. On each training and testing trial, the tester pushed a 20 cm diameter plastic bowl out through one of the doors in the screen while the handler held the dog in the holding area. When placed at the P location, the tester baited the bowl with a single piece of Royal Canin Energy® treat (1.5 cm in diameter, 30 x 50 g per sachet), this was used by the charity as a high-value reward for the dogs' regular training). In the N location and the ambiguous locations, the bowl remained empty. However, for all repetitions during training and testing, the tester placed a treat in the bowl to produce the sound of the baiting event and later gently removed it if the bowl should be empty.

When the bowl was in position, the tester stood behind the middle of the screen, facing away from the dog in front of the monitor. The tester said "Ready", and the handler released the dog with their standard "Go" cue. However, the handler was told not to encourage the dog any further. The tester timed the dog's latency to reach the bowl with a stopwatch. Each trial was finished by the tester saying "Stop" when the dog reached within 10 cm of the bowl or after 30 seconds from the dog's release, if they failed to approach the bowl. Then, the handler recalled the dog or took it back to the holding area for the subsequent trial. The tester recorded each latency in real time in a Microsoft Excel sheet modified from Hale [55].

The JBT involved two phases:

**Training phase.** Dogs were trained to approach from a starting location and discriminate when a food bowl was in a 'Positive' (P) baited (food) or a 'Negative' (N) (empty) associated location. The initial training consisted of a minimum of 15 trials. The original dog JBT [5] allowed a maximum of 50 attempts to achieve learning. However, this was reduced to 35 to avoid stressing or exhausting the dogs with overlong procedures. The training was considered complete when the dog had shorter latencies for three consecutive P presentations than each of the previous three N trials. Dogs were withdrawn from the test after the maximum number of trials if they had not met the training criteria, or if the trainer judged them to be overstressed or fatigued.

Each training session started with two bowl presentations in the P location, followed by two in the N location. The subsequent training trials interspersed both locations pseudo-randomly (positions already set in the recording sheet with an equivalent number of both N and P). There were never more than two consecutive presentations at each bowl location (Fig 3).

**Testing phase.** After the dogs reached the training criterion, they progressed immediately onto testing. An empty bowl was positioned at one of three ambiguous or probe locations between the P and N locations: 'Near Positive' (NP), 'Middle' (M) and 'Near Negative' (NN) to investigate how rapidly dogs approached the location as a measure of their responses to ambiguous cues. There were three trials for each probe location (nine in total) in the following order: M, NP, NN, NP, NN, M, NN, M, NP, with two P and two N training trials in between each probe cue to maintain the same reinforcement schedule as applied in [5]. These trials had a pseudo-random order also set in the recording sheet. In total, there were 41 test trials and a final trial with an empty P bowl to check whether the dogs were using odour to locate food. All latencies were collected during the test and subsequently analysed statistically. The procedure was stopped if a dog exceeded the maximum of 30 sec without approaching the bowl for more than five sequential trials independently of the location (Fig 4).

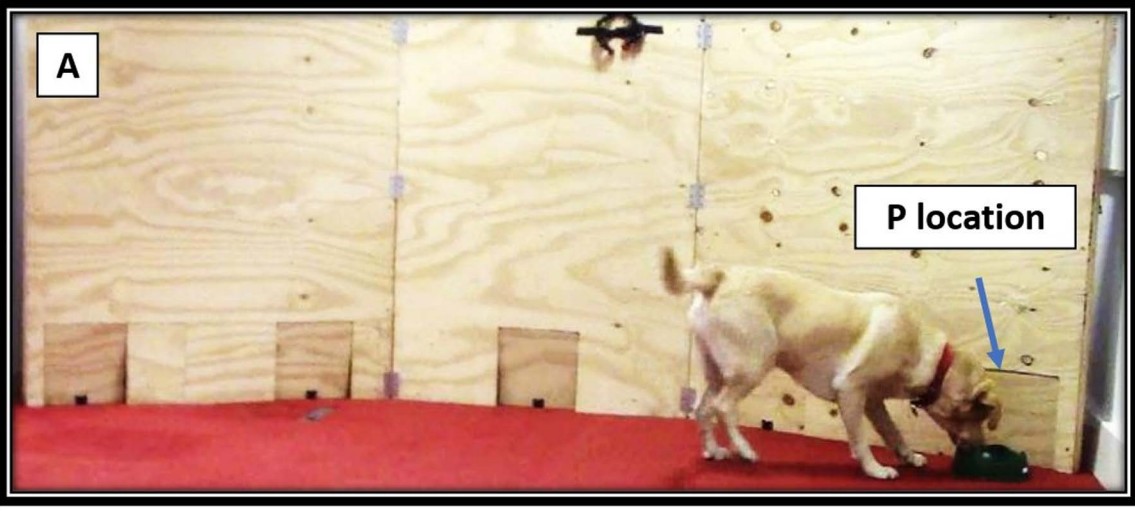

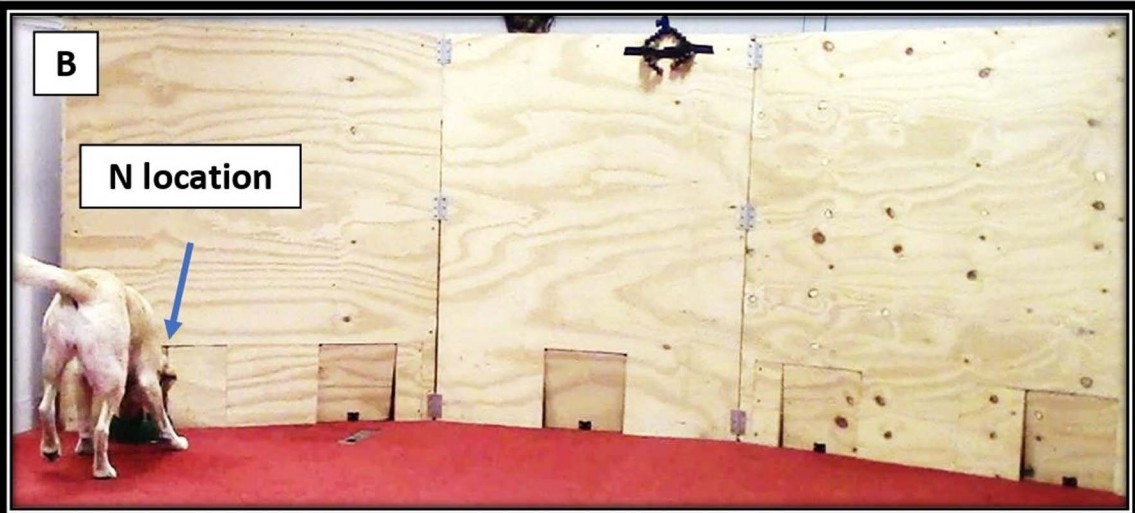

**Fig 3. JBT dog training.** Dog being trained to associate the Positive' P' location with a food reward (A) and the Negative location 'N' with the absence of reward (B). Locations were on opposite sides for half of the dogs.

### Measures of dog success in MDD tasks

We derived a range of subjective and objective MDD performance measures to quantify the dogs' success in their tasks at distinct training stages:

a) Training outcomes. We recorded whether each of the trainee dogs (N = 39) remained in the system or had failed training by the end of the experimental period. Of the trainee dogs (N = 23), 59% remained in the system at the end of data collection, whilst 41% (N = 16) were rejected.

b) Composite Total Ability Score (CTAS). We calculated a systematic subjective measure of dog performance for all dogs (N = 58) from their trainers' ratings on 27 behavioural traits considered important for MDD performance [56], based on their overall impression of the dog's performance, using a rating scale from 1 (Very low) to 5 (Very high) for each trait, and the dogs' overall task ability on a scale from 1 (one of the worst dogs I have ever seen) to 10 (one of the best dogs I have ever seen).

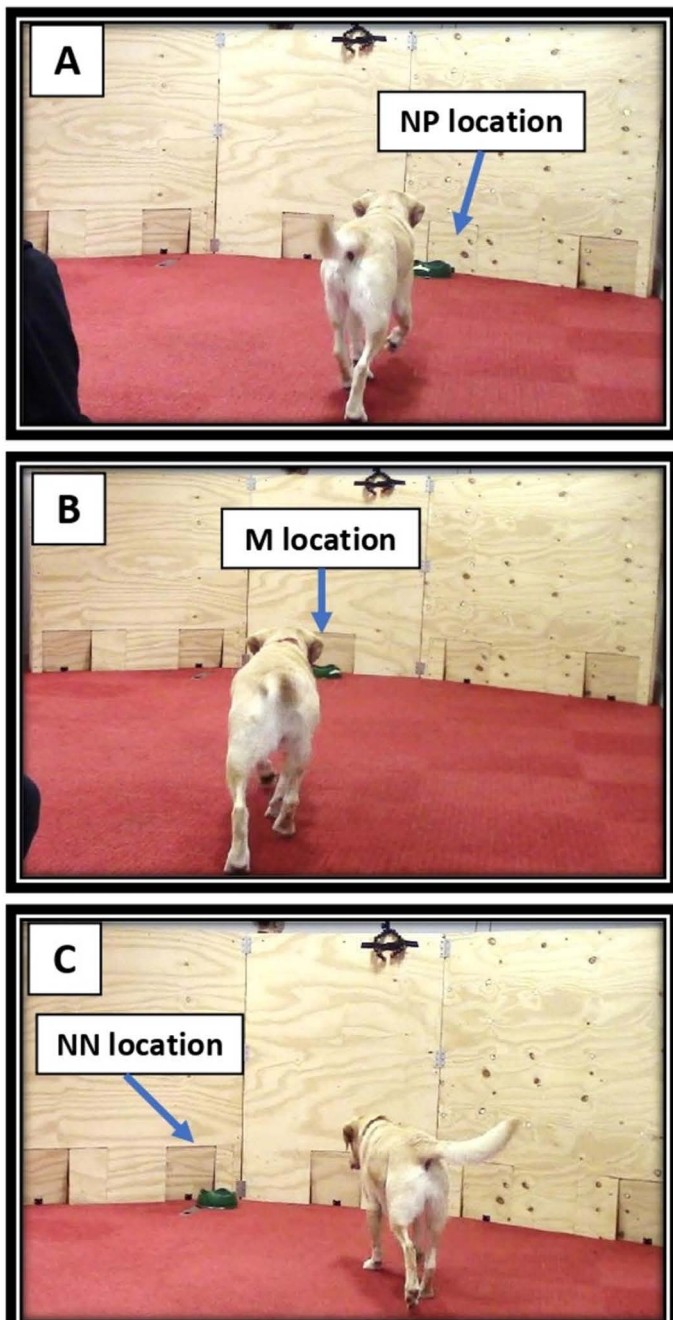

**Fig 4. JBT dog testing.** Measuring latencies to approach ambiguous locations Near positive (NP), Middle (M) and Near negative (NN).

c) Scent sensitivity and specificity scores. We collected the scores from 27 trained dogs. Although each dog was assessed on a single scent, and eight scents were represented (Table A in S1 Appendix), sensitivity and specificity scores were fairly similar across the range of conditions. The mean scent sensitivity score was 79%, varying from 50% to 100% between conditions and the mean specificity was 80%, ranging from 70% to 100% (Table

A in S1 Appendix). Therefore, we used the most recent trial data, preferably from testing trials. However, for dogs still in training, we used data from the most advanced training stage (Table B in S1 Appendix).

## Evaluation of behavioural traits

We assessed each dog using a test battery [54]. This measured the traits deemed most important to MDD performance according to a survey of professionals in the field [56]. The test battery has two parts and includes 18 cognitive and temperament subtests based on previously validated dog assessments. Subjects' (N = 58) performance in each was measured with scoring and rating scales. The test produced 98 variables. After Principal Component Analysis (PCA), these were clustered into 11 components: Component 1 'Playfulness', 2 'Persistence', 3 'Reactivity in holding pen', 4 'Food orientation', 5 'Obedience', 6 'Level of attention to handler', 7 'Self-control', 8 'Confidence', 9 'Success at problem-solving', 10 'Interest in exploring environment' and 11 'Success in search'. A summarised protocol of the test battery is provided in S2 Table. The test battery is described in detail in [54].

## Data analysis

All dogs achieved the training criterion, and all completed the judgement bias test except for two: one stopped approaching the bowl after six probe trials, thereby exceeding the maximum of 30 sec for more than five successive trials. The other dog achieved eight probe trials but reacted aggressively towards their handler when taking it back to the holding area. Both dogs were excluded from further analysis.

The data were analysed statistically using IBM® SPSS® software. The data were not normally distributed according to Shapiro-Wilk normality tests. Therefore, non-parametric tests were performed for the majority of the analyses. Regression models were conducted to reduce multiple testing when investigating the effects of multiple predictor variables on outcome measures. Although this approach may not generally be best suited to non-parametric variables, it is useful for exploratory examination of the associations between dogs' JBT performance and MDD success and test battery measures.

## Training phase analysis

The mean latency to reach the food bowls was measured for the last three training trials for P and N locations separately. The criterion for categorising a no-go response was > 30s. The difference between P and N mean latencies was calculated to assess the degree to which each dog discriminated between locations after training. We performed Spearman's Rank correlations to investigate if the difference between mean N and P latency for each dog was linked with the number of sessions they took to reach the discrimination criterion.

## Testing phase analysis

**Differences in latencies between locations.** We quantified the mean latency to approach the food bowls for each position (P, NP, M, NN and N) during testing. Friedman tests assessed differences in mean latencies to the different bowl locations. Post-hoc pairwise Wilcoxon tests, with Bonferroni corrections examined differences between each of the bowl locations.

Individual differences between dogs in size, age and breed can affect running speed. Therefore, we calculated an adjusted latency score controlling for the dogs' mean latencies to P and N locations during the testing phase. We used the following formula [5]:

$$\text{Adjusted score} = \frac{\left(\text{mean latency to probe location} - \text{mean latency to positive location}\right)}{\left(\text{mean latency to negative location} - \text{mean latency to positive location}\right)} \times 100$$

**Effect of scent cues on dogs' latencies to approach locations.** To rule out the possibility that dogs' approaches to bowls were influenced by scent cues rather than spatial discrimination learning [5], we performed an additional P trial at the end of the testing phase with an empty bowl. We compared the latency during this with the dogs' mean speed to the P location during the testing phase using a Wilcoxon signed-rank test.

**Association between dogs' behavioural and demographic characteristics and JBT measures.** We assessed the association between dogs' JBT measures and MDD behavioural and performance measures using multiple regression models to reduce multiple comparisons. We used the adjusted latency to each probe location in separate models since each might provide different information on the subjects' responses to ambiguity: dogs may interpret ambiguous locations based on their proximity to P and N trained locations as likely to be either rewarding (predicting a positive outcome, e.g., NP), or punishing (predicting absence of a positive outcome, e.g., NN). If so (though see [4] for caveats to this assumption), it is possible that variation in responses to NP may map more closely to sensitivity in detection tasks, whilst those to NN may map better to specificity metrics.

We focussed on the main effect of each independent variable on the dependent variable not on interactions; testing for associations, whilst not assuming causality. The independent variables were introduced into repeated models using stepwise methods. We included in the final model the most significant (or close to significant) independent variables that increased the overall significance of the model and the adjusted coefficient of determination ($R^2$) [57–59]. The significance level was established at $P < 0.05$. Only significant associations are mentioned in the results and discussed. However non-significant tendencies ($P < 0.10$) are also shown in the tables, as each accounts for part of the variation in the dependent variable.

For all dogs (N = 56), we assessed the association between behaviour and task performance and JBT measures. Five models assessed each JBT measure separately as the dependent variable. As independent variables, each model included dogs' demography (i.e., sex, age, training status), CTAS, and the 11 behavioural components from the test battery.

**Association between JBT measures and trainee dogs' prospective discipline and training outcomes.** For the trainee dogs (N = 39), we assessed if their JBT measures varied across disciplines and training outcomes. The models included training outcomes (whether dogs remained in the system or failed training) and prospective task (whether dogs were intended for assistance or biodetection tasks) as independent variables and each JBT measure as the dependent variable.

**Association between dogs' JBT measures and trained dogs' scent sensitivity and specificity scores.** We examined if the trained dogs' (N = 25; all dogs with scent sensitivity data except two that did not complete the test) scent sensitivity and specificity in their regular detection tasks were related to their JBT measures. The models included scent sensitivity and specificity scores as independent variables and each JBT measure as the dependent variable.

## Results

### Training phase

The dogs took an average of 18.42 ± 4.4 trials (min = 15, max = 35) to reach the training criterion and progress to the testing phase. The mode was 15 trials; only two dogs took > 30 trials to achieve discrimination criteria. Mean latencies to P ranged from 1.5 to 4.5 sec (mean = 2.46

± 0.59) and to N from 2.3 to 29.3 sec (mean = 8.25 ± 6.78). The mean difference between mean P and N latencies was 5.8 ± 6.5 sec (min = 0.26, max = 26.91). The difference between mean P and N latencies was significantly negatively associated with the number of trials to achieve the training criterion (R = -0.437, p = 0.001), showing dogs reaching the training criterion faster showed greater discrimination between N and P locations in the training phase.

### Testing phase

**Variation in individual latencies among dogs.** Dogs showed individual variability in the average time taken to reach each location. For instance, they exhibited a broad range of latency to reach the P (overall mean = 3.36 ± 2.24, min = 1.64, max = 16.49) and N (overall mean = 17.60 ± 6.88, min = 3.05, max = 28.93) locations during testing.

**Differences in latencies between locations.** The latency to reach the bowl was significantly affected by location (Friedman's test: N = 56, $\chi^2(2)$ = 176.57, $p < 0.001$), with significant differences between each of the bowl locations, except for P vs NP (Fig 5).

**Effect of scent cues on dogs' latencies.** When comparing the dogs' latency to approach P with an empty bowl (mean = 2.93 sec ± 0.8) and the mean latency towards P during the testing phase (mean = 3.15 sec ± 1.41), no significant difference was found (z = -0.2, p = 0.842), suggesting that there was no influence of odour cues on the dogs' speed to approach locations.

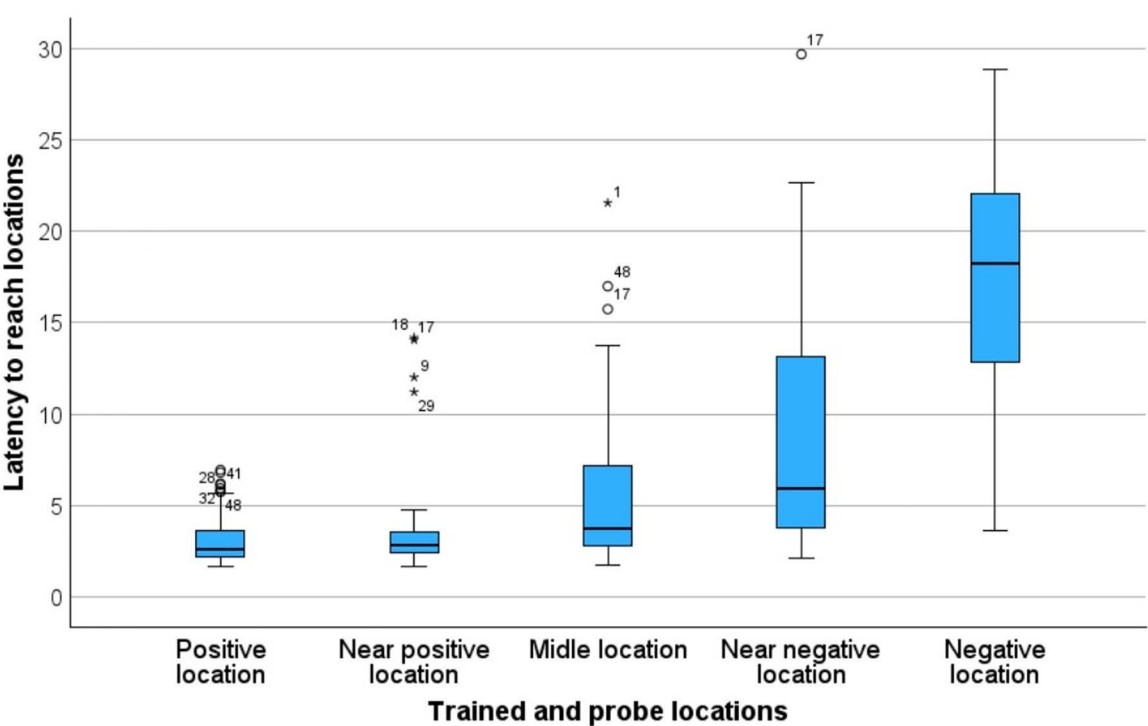

**Fig 5. Distribution of mean latencies to approach each bowl location during testing (in secs) (N = 56).** Boxes show the median (bar within the box), the 25th interquartile (lower box border) and the 75th interquartile (upper box border). The whiskers indicate the minimum and maximum mean latencies, and the circles and stars represent outliers.

## Exploratory regression analyses of associations between dogs' behavioural and demographic characteristics and JBT measures

**Near Positive (NP) adjusted latencies scores.** For the NP location, the regression model was overall significant ($F_{(3, 52)}$ = 6.435, p < 0.001, adj. R2 = 23%). Older dogs, and dogs with higher 'Confidence' scores were quicker to approach the NP location (Table 1 and Fig 6a).

**Middle (M) adjusted latencies scores.** The analysis derived a significant model ($F_{(7, 48)}$ = 4.377, p < 0.01, adj. R2 = 24%), indicating that dogs with a higher Composite Total Ability Score (CTAS), those with higher 'Playfulness' and those with higher 'Food orientation' tended to approach the M location significantly faster (Table 2 and Fig 6b,c,d). The association with the remaining factors was not significant.

**Near Negative (NN) adjusted latencies scores.** There were no significant associations between NN adjusted latency score and dogs' demography, behavioural characteristics or outcome measures.

## Exploratory regression analyses of associations between dogs' JBT measures, training outcome, task type and sensitivity and specificity

Neither the trainee dogs' training outcome nor their prospective task was significantly linked to JBT measures.

Only the model assessing the link between trained dogs' sensitivity and specificity scores and latency to M was significant ($F_{(1, 23)}$ = 6.126, p < 0.05, adj. R2 = 18%). Dogs that moved

**Table 1. Association between the dogs' demography and behavioural characteristics (N = 56) and their latencies to approach NP. ***P < 0.001.**

| Independent variables | ß | 95.0% CI for B | | SE B | P | β | R2 | ΔR² |
|---|---|---|---|---|---|---|---|---|
| | | LL | UL | | | | | |
| Model | | | | | | | 0.27 | 0.23*** |
| Constant | 10.35 | 5.30 | 15.40 | 2.52 | <0.001 | | | |
| Age in months | -0.23 | -0.35 | -0.10 | 0.06 | <0.001 | -0.45 | | |
| Confidence | -4.34 | -7.54 | -1.14 | 1.60 | 0.009 | -0.33 | | |
| Success at problem solving | -2.60 | -5.70 | 0.50 | 1.54 | 0.098 | -0.20 | | |

Note. B, unstandardised regression coefficient; CI, confidence interval; LL, lower limit; UL, upper limit; SE B, standard error of the coefficient; β, standardised coefficient; R², coefficient of determination; ΔR², adjusted R2.

**Table 2. Association between the dogs' demography and behavioural characteristics (N = 56) and their latencies to approach M **P < 0.01.**

| Independent variables | B | 95.0% CI for B | | SE B | P | β | R2 | ΔR² |
|---|---|---|---|---|---|---|---|---|
| | | LL | UL | | | | | |
| Model | | | | | | | 0.34 | 0.24** |
| (Constant) | 48.10 | 29.24 | 66.96 | 9.38 | <0.001 | | | |
| CTAS | -0.53 | -0.85 | -0.20 | 0.16 | 0.002 | -0.44 | | |
| Playfulness | -10.11 | -17.07 | -3.15 | 3.46 | 0.005 | -0.37 | | |
| Food orientation | 8.42 | 1.75 | 15.09 | 3.32 | 0.014 | 0.31 | | |
| Obedience | 6.04 | -0.59 | 12.67 | 3.30 | 0.073 | 0.23 | | |
| Level of attention to handler | -6.05 | -13.16 | 1.06 | 3.54 | 0.093 | -0.22 | | |
| Success in search | 5.46 | -0.98 | 11.90 | 3.20 | 0.095 | 0.21 | | |

Note. B, unstandardised regression coefficient; CI, confidence interval; LL, lower limit; UL, upper limit; SE B, standard error of the coefficient; β, standardised coefficient; R², coefficient of determination; ΔR², adjusted R2.

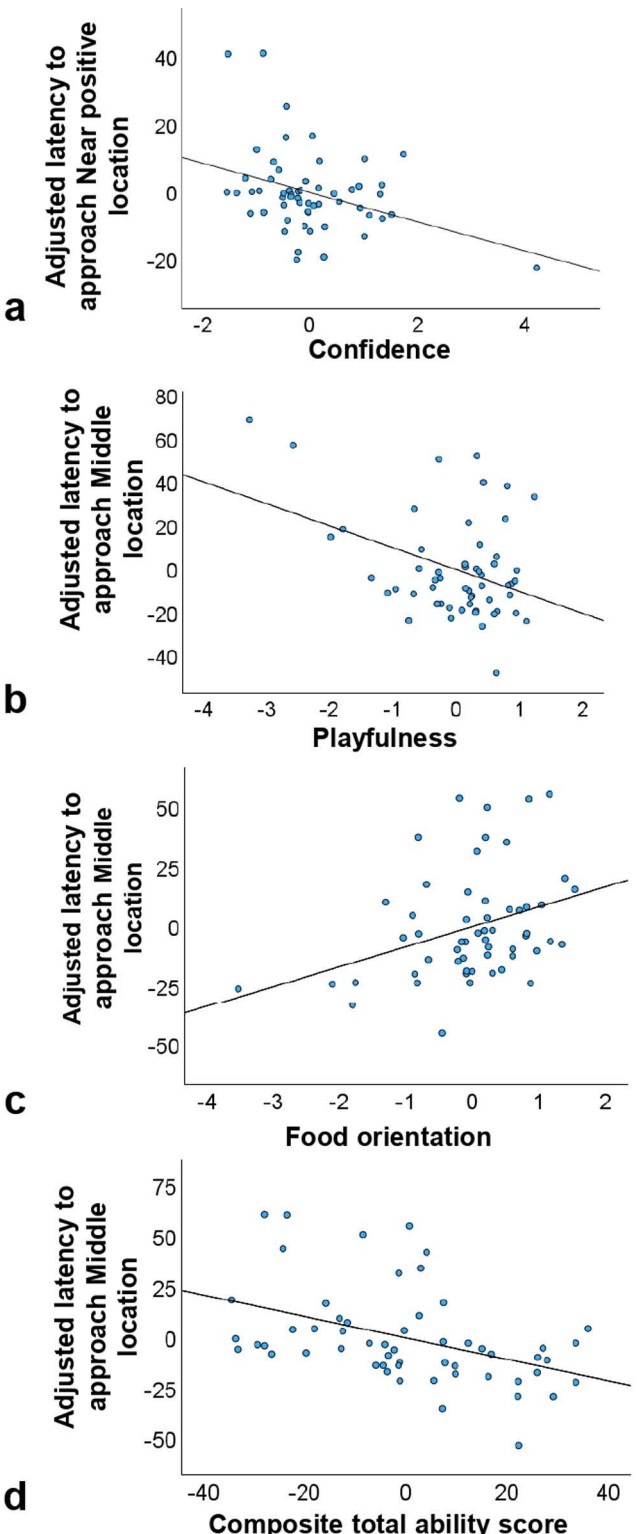

**Fig 6. Associations between dogs' behavioural characteristics and adjusted JBT measures (N = 56).** The charts display significant correlations from the mixed models between behavioural components from the test battery and adjusted latencies towards (a) near positive and (b,c) middle locations, and the association between (d) CTAS and middle location.

more slowly towards M tended to have higher scent specificity in their detection tasks (Table 3, Fig. 7). However, there was no significant relationship between sensitivity and JBT measures.

## Discussion

### Is the outcome of JBT associated with dogs' performance in MDD tasks?

This study investigated the relationship between the behaviour and performance of MDDs and their performance in a JBT. All dogs learned the task during training and achieved generalisation in their response across the different locations. Most dogs completed the test (only two out of 58 were excluded). This contrasts with previous studies reporting high levels of dog exclusion [e.g., 18,55,60]. Dogs with higher Composite Total Ability Scores (CTAS) approached the ambiguous Middle (M) location faster, whilst those with higher scent specificity in detection tasks took longer to approach M. Some demographic and test battery measures were also associated with the dogs' behaviour in JBT. Older dogs and those with higher 'Confidence' approached the ambiguous NP location faster whilst dogs with higher 'Playfulness' and higher 'Food orientation' approached the M location faster.

**Table 3. Association between trained dogs' scent specificity scores (N = 25) and their latency to approach M **P < 0.01.**

| Independent variables | B | 95.0% CI for B | | SE B | P | β | R2 | ΔR² |
|---|---|---|---|---|---|---|---|---|
| | | LL | UL | | | | | |
| Model | | | | | | | 0.21 | 0.18* |
| (Constant) | -129.73 | -257.71 | -1.76 | 61.86 | 0.047 | | | |
| Scent specificity | 171.57 | 28.17 | 314.97 | 69.32 | 0.021 | 0.46 | | |

Note. B, unstandardised regression coefficient; CI, confidence interval; LL, lower limit; UL, upper limit; SE B, standard error of the coefficient; β, standardised coefficient; R², coefficient of determination; ΔR², adjusted R2.

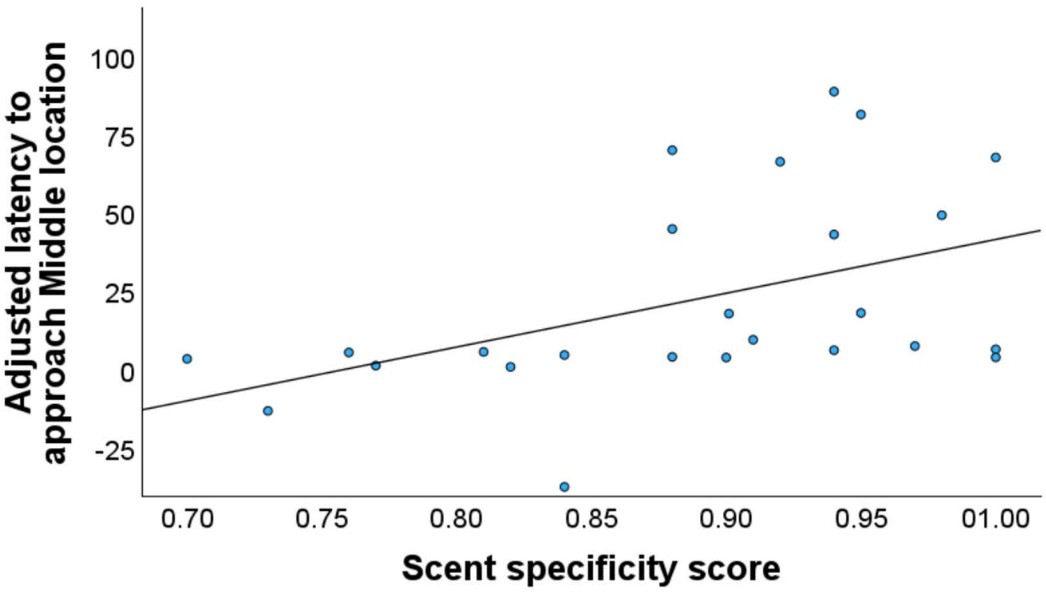

**Fig. 7. Association between the dogs' scent specificity scores and adjusted JBT measures.** Significant correlation between scent specificity scores for trained dogs (N = 25) and their adjusted latency to approach M location.

**Association between dog's performance in JBT with measures of success in MDD tasks.** The finding that dogs more likely to judge the M ambiguous location as positive also scored higher on CTAS in the MDD task could be due to some aspect of 'optimistic' decision-making being associated with good performance in detection tasks, a more positive affective state (as inferred from JBT) being linked with good performance. For example, a more 'optimistic' dog (as inferred from 'optimistic' responses) might have a higher motivation to engage with tasks due to expecting positive outcomes [61,62]. Also, a positive attitude might enhance learning and memory processes [21,62–64], making it easier for 'optimistic' dogs to remember and apply training from detection tasks more effectively and hence receive a high CTAS during training or operating in detection tasks. However, a more negative affective state (as inferred from 'pessimistic' responses) may predispose an adaptive tendency to save energy if resources are insufficient or when goal-oriented efforts (e.g., during training) are repeatedly unsuccessful [65], potentially increasing the chances of disengagement from the task and hence poor CTAS ratings. To parse these different types of explanation, further research could incorporate other putative markers of affective state (e.g., physiological indicators; psychometric scales such as the PANAS [66] or Canine Frustration Questionnaire [67]), to allow independent evaluation of the links between affect and performance that may help to differentiate these from possible links between decision-strategy and performance.

There was no significant association between JBT performance and training outcome in the trainee dogs. Hence JBT may not predict whether they pass or fail training. The binary nature of the training outcome measure may be a less sensitive parameter of aptitude than other graded measures (e.g., CTAS), and any subtle link may not be detectable in this relatively small sample size.

The association between 'optimistic' responding to the M location and lower scent specificity indicates a link between an individual's expectations of reward when judging spatial ambiguity in JBT and its propensity to make a positive response under perceptually noisy conditions in a separate odour detection task. One possibility is that general risk-proneness, impulsivity, and/or elevated valuation of reward underpins both measures [68–73]. In line with the latter possibility, we also found an association between higher 'food orientation' and 'playfulness' and 'optimistic' responding in JBT (see later).

From an applied perspective, JBT tests may thus be useful in detecting 'optimistic' dogs who are also more 'liberal' in their odour detection criteria than more 'pessimistic' dogs who are more 'conservative' [38] and perhaps weigh up their decision to alert to a scent for longer before acting. The link between higher specificity and pessimistic responses supports this. Because MDDs must discriminate scents at small concentrations, and it is desirable to minimise erroneous indications in either direction, screening of potential trainees on a range of tests, perhaps including JBT, could maximise efficiency and would be worth studying further.

**Association between demography, dogs' behavioural measures and JBT measures.** The only demographic factor associated with JBT measures was age. The relationship between increased age and shorter latencies to NP is somewhat unexpected. It was anticipated that younger dogs, potentially more curious and energetic, would run faster toward the bowl [74]. However, older dogs may have had more training and experience, thus persevering in their task [32]. In contrast, younger dogs may approach faster initially but may be less focused, becoming frustrated sooner when not rewarded and cease running to the probe locations. Alternatively, younger dogs may be faster to learn that ambiguous cues are not rewarded. In a between-subjects JBT study, Piotti et al. [20] compared the performance of older and younger pet dogs, hypothesising that older dogs would be more pessimistic towards ambiguous stimuli. However, the speed to approach ambiguous probes did not vary significantly with age.

**Association between dogs' behavioural measures in the test battery and JBT measures.** The finding that dogs with higher 'Confidence' were more 'optimistic' in the JBT (at the NP location) aligns with suggestions that higher confidence is associated with more positive, active responses to different situations [75]. Similar findings come from a study assessing the relationship between performance in a JBT and personality traits from the Boldness-Shyness continuum and the C-BARQ questionnaire; dogs that were more sociable and excitable approached ambiguous locations faster whilst those with higher separation-related behaviours and aggressiveness/ fearfulness towards other dogs took longer to approach [76]. Furthermore, dogs with fearful tendencies and anxiety-related disorders tended to have longer latencies towards probe locations [5,13].

There are also parallels with the human psychology literature which describes optimistic individuals as more emotionally stable and extroverted. They are more likely to have an adaptive outlook, facilitating problem-solving and coping mechanisms in stressful situations, and being more cognitively flexible, with better memory consolidation, and quicker decision-making. [77–79]. In contrast, anxious individuals may take longer to react to neutral stimuli, making more errors or interpreting them as more threatening than less anxious subjects e.g., [2,80,81].

Dogs with higher 'Playfulness' tended to approach the M location faster. Play behaviour is widely regarded as an indicator of animal welfare [82–85]. It is frequently associated with positive emotions and the activation of reward systems [84,86], occurring when no immediate fitness threats exist [83]. Conversely, the absence of play can indicate compromised wellbeing, a negative affective state or poor health status in animals [83,84,87]. Play has been also linked with different positive aspects in humans, such as favourable life quality [88], stress coping [89] and educational accomplishment [90]. In dogs, playfulness has been associated with better performance for highly active detection tasks [91–93]. However, excessive playfulness may come with high excitability, especially in developmentally immature dogs [67,74] and may prove disruptive in certain tasks, such as medical detection, where dogs operate in controlled environments or public spaces. Hence, an intermediate level of playfulness may be favourable for MDD performance, increasing underlying positive affective states without being task-disruptive. Dogs with a higher 'Food orientation' also ran faster towards the bowl, possibly anticipating a food reward. This positive expectation may lead reward-sensitive dogs to persist in seeking food [6].

In summary, the apparent cluster of higher 'Confidence' (which may well increase with age), 'Playfulness', 'Food orientation' and 'optimism' may predict dogs who have lower 'specificity' in detection tasks but, at the same time, are more easily trained due to a tendency to persist at tasks.

## Why are some performance measures linked with different probe locations?

The relationships between performance measures and their latencies to approach the bowl varied across probe locations. It is possible that different locations provide different information about affective states [1,6,94]. For example, ambiguous locations closer to the P location may be better at detecting variation in reward expectations (often linked to depression), whilst those near the N location may be more sensitive to variation in expectation of punishment (often linked to anxiety). However, this requires balanced valuation of the trained decision outcomes which is rarely tested, and there are other caveats (see [4], section 3.2.4). For example, here N was associated not with punishment but with the absence of a reward negating the above arguments.

In this study, most dogs had short latencies to the NP location and did not significantly differentiate between P and NP, suggesting high expectations of reward across these locations. Lack of variation between dog responses to this ambiguous location likely precluded the detection of links between NP responses and other variables. The most plausible reason for significant differences being detected at the middle rather than the other ambiguous cue locations is that, due to its central location, it presented the highest level of ambiguity to the dogs. As such, responses to it are most likely to have been sensitive to influence by individual-specific states and traits, and hence to reveal associations with related predictor variables. It is conceivable that ambiguity of the middle position was further increased by noise or other cues from the tester positioned behind this location despite measures taken to minimise such effects, including visual occlusion. However, any such unintentional cues are likely to have varied between trials and hence not to have exerted a consistent influence on dog responses.

## Are medical detection dogs different from other populations?

Previous studies with a range of dog population have had a high level of dropouts in the JBT test. For instance, in Hale [55], 41.6% of the dogs did not complete the test, possibly resulting in selective subject inclusion [6]. However, in our study no dogs failed JBT training, and only two did not complete the test. MDDs may have engaged more with JBT than other dog populations due to their training discipline, genetics, and preselection. These dogs regularly participate in training exercises for prolonged periods, and they were familiar with the experimental room and the charity's facilities. In other studies with higher levels of subject exclusion, the dogs could have experienced neophobia because the JBT was carried out in places unknown to them or because they were more anxious than MDDs and had less ability to cope with the task. When Hale [55] assessed a population of 101 fearful dogs, 42 did not complete the JBT, and of these, 25 showed visible stress-related signals. Alternatively, pet and shelter dogs may show varied energy levels, while working dogs tend to have high stamina [e.g., 75,95] and may be more able to complete the JBT without becoming fatigued compared to other populations.

MDDs learned to discriminate between P and N relatively fast, taking 18.6 trials on average. Other dog populations took longer, e.g., shelter dogs mean = 29.42 trials [5]; pet dogs mean = 42 trials [60]. It was also interesting that dogs who learned faster tended to achieve better discrimination. A similar finding was reported by Gruen [21] who observed that dogs that took longer to approach probe locations in the JBT also learned faster in spatial memory tasks. They suggested that these dogs may have been generally fast learners, being both quicker to learn the discrimination between trained and ambiguous cues in the JBT, and better at learning other spatial tasks. Perhaps JBT training achievement may be a good predictor of dogs' general learning skills in working dog populations and this may be worth exploring further, e.g., with a reversal learning task, and in other detection dog disciplines.

MDDs may be more 'optimistic' than other dog populations as they are raised under standardised socialisation and training to make them resilient across contexts and in their roles. Although their behaviour may vary across disciplines and training stages, these dogs may be more emotionally stable than companion or dogs in rehoming kennels, who are from variable origins and with wide ranging previous experiences. Comparative studies between naïve and working dog samples may provide more information on how JBT performance varies across dog populations and whether testing in naïve dogs predicts later performance after training.

## Study limitations and future steps

In this study, decision-making under ambiguity in a JBT task was associated with measures of how dogs perform MDD tasks. JBT tasks might thus be valuable predictors of MDD

performance that are low cost, require few materials, and are relatively quick to perform with data collectable in real-time (i.e., measuring latencies using a stopwatch). The JBT method used here incorporating a wooden panel provided a relatively uniform backdrop to the choice context devoid of potentially distracting room features. It also allowed the tester to manipulate the food bowls out of sight of the dogs or handler, reducing unintentional cueing of the bowl content, i.e., clever Hans effect [96,97]. However, the study also has certain limitations.

Firstly, individual variation in performance in test batteries and MDD tasks may be influenced by various variables, some of which we aimed to standardise or measure from the study's population, but others (e.g., dog (social) experiences, 'personality' traits) may be more challenging to control and hence contribute to unexplained variation. The factors underlying the associations detected between JBT, test battery and MDD performance thus remain unknown at this point. Since a number of JBT studies (e.g., rats [47]; pigs [48]; junglefowl [49]; mice [50]; cows [51]) have suggested the existence of cross-time consistency in 'optimism' or 'pessimism' which may reflect a trait component of decision-making under ambiguity that might, as indicated here, influence other aspects of the animals' behaviour, future studies could further evaluate JBT consistency across time in dogs. Studies with larger sample sizes could also investigate other factors such as training regime and experiences, target scents used, that may underlie the associations detected here.

Secondly, the test battery preceded JBT on the same day. This offered advantages as the dogs were familiar with the experimental arena and the general testing dynamics, allowing us to proceed immediately with JBT. However, it could have been exhausting for the dogs, or otherwise affected their performance, despite allowing a rest period of at least two hours. Ideally, we would have performed the JBT on a different day from the test battery. However, this was impossible due to logistical and time limitations.

## Conclusion

This exploratory study identified significant links between measures of dogs' general ability in MDD tasks and their tendency to be more 'optimistic' or 'pessimistic' in a JBT. The finding that dogs with a more 'pessimistic' response in JBT also showed greater specificity in MDD tasks raises the possibility that JBTs might be predictive of decision-tendencies in medical detection tasks. MDD performance on the JBT also showed some differences to other dog populations. MDDs were more likely to participate and persist on the test and learned faster than shelter and pet dogs in comparable studies. The utility of the test to screen potential detection dogs requires further investigation, for example by exploring whether JBT responses can help predict dogs that are rejected from training. This may facilitate the selection of well-suited dogs, overall improvements to training efficiency, and enhanced dog wellbeing.

## Supporting information

**S1 Table. MDD sample demographic details.** Including general information, training status when tested and at the data collection endpoint and training outcome.
(DOCX)

**S2 Table. Test battery Parts 1 and 2.** Subtests description, traits assessed and variables definition.
(DOCX)

**S1 Appendix. Additional information on scent sensitivity and specificity conditions and scent exposures.** Table A shows the ranges of scent sensitivity and specificity across

conditions. Table B includes the number of exposures and presented scent samples, target source, and the number of training and testing trials for each target scent.
(DOCX)

## Acknowledgements

We want to thank all the staff from Medical Detection Dogs for their invaluable help. A special thanks to Helena Hale for all her advice on the test performance and Rob Harris for his technical support, and Chris Allen for help scheduling dog use. Thanks to Sheila Brill and Ian Staniland for proofreading the thesis on which this paper is based.

## Author contributions

**Conceptualization:** Sharyn Bistre Dabbah, Michael Mendl, Nicola J Rooney.

**Data curation:** Sharyn Bistre Dabbah, Michael Mendl.

**Formal analysis:** Sharyn Bistre Dabbah, Michael Mendl.

**Funding acquisition:** Sharyn Bistre Dabbah.

**Investigation:** Sharyn Bistre Dabbah, Michael Mendl, Nicola J Rooney.

**Methodology:** Sharyn Bistre Dabbah, Michael Mendl, Nicola J Rooney.

**Project administration:** Sharyn Bistre Dabbah, Michael Mendl, Claire Guest, Nicola J Rooney.

**Resources:** Sharyn Bistre Dabbah, Michael Mendl, Claire Guest, Nicola J Rooney.

**Supervision:** Michael Mendl, Claire Guest, Nicola J Rooney.

**Validation:** Sharyn Bistre Dabbah, Michael Mendl, Claire Guest, Nicola J Rooney.

**Visualization:** Sharyn Bistre Dabbah, Michael Mendl.

**Writing – original draft:** Sharyn Bistre Dabbah.

**Writing – review & editing:** Sharyn Bistre Dabbah, Michael Mendl, Claire Guest, Nicola J Rooney.

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
