## [Decision Letter · Decision Letter 0]

18 Dec 2024

PONE-D-24-49123An exploratory study of associations between judgement bias, demographic and behavioural characteristics, and detection task performance in medical detection dogsPLOS ONE

Dear Dr. Rooney,

Thank you for submitting your manuscript to PLOS ONE. After careful consideration, we feel that it has merit but does not fully meet PLOS ONE’s publication criteria as it currently stands. Therefore, we invite you to submit a revised version of the manuscript that addresses the points raised during the review process. We are so sorry but we could obtain the comments from only one reviewer.

We look forward to receiving your revised manuscript.

Kind regards,

Etsuro Ito, Ph.D.

Academic Editor

PLOS ONE

Journal Requirements:

“I have read the journal's policy and the authors of this manuscript have the following competing interests: CG and NR are employed by Medical Detection Dogs, the charity where the dogs from this study are trained and data collection was conducted.  However, they were not involved in data collection or analysis for this study.”

Please confirm that this does not alter your adherence to all PLOS ONE policies on sharing data and materials, by including the following statement: "This does not alter our adherence to  PLOS ONE policies on sharing data and materials.” (as detailed online in our guide for authors http://journals.plos.org/plosone/s/competing-interests).  If there are restrictions on sharing of data and/or materials, please state these. 

Please note that we cannot proceed with consideration of your article until this information has been declared. 

Reviewers' comments:

Reviewer's Responses to Questions

**Comments to the Author**

1. Is the manuscript technically sound, and do the data support the conclusions?

Reviewer #1: Yes

2. Has the statistical analysis been performed appropriately and rigorously? 

Reviewer #1: Yes

3. Have the authors made all data underlying the findings in their manuscript fully available?

Reviewer #1: Yes

4. Is the manuscript presented in an intelligible fashion and written in standard English?

Reviewer #1: Yes

5. Review Comments to the Author

Reviewer #1: The manuscript is well-written and organized, following a clear roadmap of events. The introduction provides a compelling rationale for the study, and the methods are described with sufficient detail, allowing for reproducibility.

The statistical analyses are robust and thoroughly explained, and the results are presented in a logical sequence that mirrors the research questions, making it easy to follow the progression of the study.

This is a very interesting and innovative piece of work that explores the relationship between personality traits and decision-making in medical detection dogs. The findings provide valuable insights into how affective and cognitive processes might influence detection performance. By identifying traits such as optimism, confidence, and behavioral tendencies that correlate with task outcomes, the study opens the door to further research in this area.

Additionally, the study highlights the potential utility of judgment bias tasks as predictive tools, which could have implications for the selection and training of other types of detection dogs. The work has significant practical and theoretical implications, making it a strong contribution to the field.

I have no major concerns, but I would encourage the authors to consider:

The authors should consider discussing alternative explanations for why the middle position was significantly selected by the dogs. While the findings suggest that more playful and food-motivated dogs approached the middle position faster, interpreted as optimism, the association between higher specificity and slower approaches to the same position suggests a more cautious, discriminative behavior. This raises questions about the underlying factors influencing middle position selection. For instance, could the middle position's significance be related to training protocols, novelty of the setup, or the fact that the tester was positioned behind this area? It is possible that dogs heard subtle cues from the tester or other background noise, leading them to anticipate something about the task. Exploring whether this position inherently holds some predictive or environmental cues could provide deeper insights into the observed behaviors and refine interpretations of optimism versus discrimination tendencies.

Overall, this is an interesting study with the potential to inspire further research in both medical and other detection dog applications.

6. PLOS authors have the option to publish the peer review history of their article (what does this mean? ). If published, this will include your full peer review and any attached files.

**Do you want your identity to be public for this peer review?** For information about this choice, including consent withdrawal, please see our Privacy Policy .

Reviewer #1: **Yes: ** Astrid R. Concha

---

## [Author Response · Author response to Decision Letter 1]

11 Feb 2025

Thank you for this interesting comment. In our view, and as outlined in lines 610-614 of the original paper, the most plausible reason for significant differences being detected at the middle rather than the other ambiguous cue locations is that, due to its central location, it presented the highest level of ambiguity to the dogs. As such, responses to it are most likely to have been ‘vulnerable’ to influence by individual-specific states and traits, and hence to reveal associations with related predictor variables. It is perhaps also possible that ambiguity of the middle position was further increased by noise or other cues from the tester positioned behind this location despite measures taken to minimise such effects, including visual occlusion. Any such unintentional cues are likely to have varied between trials and hence not to have exerted a consistent influence on responses. We have edited the text to include these points (Line 610-619 Revised Manuscript Clean).

---

## [Editor Report · Decision Letter 1]

14 Feb 2025

An exploratory study of associations between judgement bias, demographic and behavioural characteristics, and detection task performance in medical detection dogs

PONE-D-24-49123R1

Dear Dr. Rooney,

We’re pleased to inform you that your manuscript has been judged scientifically suitable for publication and will be formally accepted for publication once it meets all outstanding technical requirements.

Kind regards,

Etsuro Ito, Ph.D.

Academic Editor

PLOS ONE

---

## [Editor Report · Acceptance letter]

PONE-D-24-49123R1

PLOS ONE

Dear Dr. Rooney,

I'm pleased to inform you that your manuscript has been deemed suitable for publication in PLOS ONE. Congratulations! Your manuscript is now being handed over to our production team.

Kind regards,

on behalf of

Prof. Etsuro Ito

Academic Editor

PLOS ONE